# Mutations in Genes with a Role in Cell Envelope Biosynthesis Render Gram-Negative Bacteria Highly Susceptible to the Anti-Infective Small Molecule D66

**DOI:** 10.3390/microorganisms13071521

**Published:** 2025-06-29

**Authors:** Samual C. Allgood, Calvin A. Ewing, Weiping Chu, Steffen Porwollik, Michael McClelland, Corrella S. Detweiler

**Affiliations:** 1Department of Molecular, Cellular and Developmental Biology, University of Colorado, Boulder, CO 80309, USA; samual.allgood@colorado.edu (S.C.A.); calvin.ewing@colorado.edu (C.A.E.); 2Department of Microbiology and Molecular Genetics, University of California, Irvine, Irvine, CA 92697, USAsporwoll@uci.edu (S.P.); mmcclell@uci.edu (M.M.)

**Keywords:** AcrAB-TolC, anti-infective, efflux pump, *gmhB/yaeD*, inner membrane, lipopolysaccharide (LPS), outer membrane, Resistance Nodulation cell Division (RND), salmonella, *ygeG*

## Abstract

Anti-infectives include molecules that target microbes in the context of infection but lack antimicrobial activity under conventional growth conditions. We previously described D66, a small molecule that kills the Gram-negative pathogen *Salmonella enterica* serovar Typhimurium (*S.* Typhimurium) within cultured macrophages and murine tissues, with low host toxicity. While D66 fails to inhibit bacterial growth in standard media, the compound is bacteriostatic and disrupts the cell membrane voltage gradient without lysis under growth conditions that permeabilize the outer membrane or reduce efflux pump activity. To gain insights into specific bacterial targets of D66, we pursued two genetic approaches. Selection for resistance to D66 revealed spontaneous point mutations that mapped within the *gmhB* gene, which encodes a protein involved in the biosynthesis of the lipopolysaccharide core molecule. *E. coli* and *S.* Typhimurium *gmhB* mutants exhibited increased resistance to antibiotics, indicating a more robust barrier to entry. Conversely, *S.* Typhimurium transposon insertions in genes involved in outer membrane permeability or efflux pump activity reduced fitness in the presence of D66. Together, these observations underscore the significance of the bacterial cell envelope in safeguarding Gram-negative bacteria from small molecules.

## 1. Introduction

Gram-negative bacteria have a cell envelope that is a highly effective protective barrier against environmental threats, including antibiotics, detergents, and host defense molecules [1]. The cell envelope includes an asymmetric outer membrane composed of an inner leaflet of phospholipids and an outer leaflet of densely packed lipopolysaccharides (LPS) [2]. The Lipid A component of LPS is stabilized by divalent cations (Mg^2+^/Ca^2+^), creating a rigid, low-fluidity surface that impedes the entry of hydrophobic small molecules [3,4]. Also within the cell envelope, tripartite Resistance-Nodulation cell Division (RND) efflux pumps, such as AcrAB-TolC, traverse the inner membrane, the periplasm, and the outer membrane [5]. RND efflux pumps capture substrates from the periplasm or the inner membrane and expels them from the cell, thereby minimizing the accumulation of small molecules in the cytosol, inner membrane, and periplasm [6].

Numerous studies have demonstrated that conditions compromising the cell envelope increase bacterial sensitivity to antibiotics and other small molecules [7,8,9]. In growth media that reduce outer membrane integrity or efflux pump activity, bacteria are more susceptible to antibiotics that are normally excluded by the cell envelope. For instance, mutations in LPS biosynthesis genes or in subunits of the AcrAB-TolC efflux pump increase sensitivity to the antibiotic novobiocin [10,11]. Moreover, compounds or antibiotics that inhibit Gram-negative bacterial growth under conditions that compromise the cell envelope show anti-infective activity in host cells and whole animals [12,13,14,15]. For these reasons, experiments in host cells or under laboratory conditions that mimic soluble host physiology have become popular tools for identifying small molecules that inhibit Gram-negative bacteria [16,17,18,19]. Such approaches identify compounds that directly target bacteria even when the molecules are ineffective in standard media due to an intact LPS layer and/or functional RND efflux pumps [18,20,21]. These observations suggest that during infection, accumulated damage to the LPS layer or occupancy of RND efflux pumps enhances small-molecule access to bacterial cell targets.

The bacterial inner or cytoplasmic membrane is an essential structure that supports a complex set of interrelated, critical processes, including nutrient uptake, response to stress, export of waste and/or toxins, energy generation, signaling, and the anchoring of key bacterial structures, such as secretion systems, the cell wall, and surface appendages [22]. Many of these processes contribute to pathogen survival during infection. Unfortunately, compounds that damage bacterial cell membranes often lyse them, raising the specter of toxicity [21,23,24]. However, small molecules have been described that, in host cells and whole animals, attenuate bacterial infection and, in broth culture, disrupt membrane voltage without causing overt bacterial cell lysis [25,26]. Understanding the mechanisms of action of such compounds may reveal compound characteristics and activities desirable in new anti-infectives.

Previously, a small hydrophobic molecule, D66 (378 g/mol; cLogP 4.73), was shown to enable the killing of *S.* Typhimurium within cultured macrophages and in mice. However, in broth culture, D66 lacks anti-bacterial activity unless the cell envelope barrier is compromised, in which case the compound rapidly increases membrane fluidity and disrupts voltage (*∆*ψ) without lysis of the bacterial cell membrane [25,27]. For instance, exposure of *S.* Typhimurium to a cationic antimicrobial peptide and D66 in Lysogeny Broth (LB) resulted in ejection of the fluorescent probe 3,3′-dipropylthiadicarbocyanine iodide [DiSC3(5)] from the cell membrane while preventing propidium iodide access to the DNA [25]. The cationic antimicrobial peptide utilized was polymyxin B at 0.5 μg/mL, a concentration empirically determined to permeabilize the outer membrane LPS barrier but not the inner membrane [20]. The polymyxin B nonapeptide derivative did not potentiate D66 at concentrations (20 ug/mL) that enable novobiocin to reach cellular targets [25,28]. These observations could suggest that PMB potentiates D66 not only by permeabilizing the outer membrane but also via the generation of reactive oxygen species [28,29]. A complementary approach supported the idea that the cell envelope barrier excludes D66. The AcrAB-TolC RND efflux pump plays a major role in protecting Enterobacterales from small, toxic molecules [29]. An *S.* Typhimurium strain lacking *acrAB* and an *E. coli* strain lacking *tolC* were more sensitive to D66 than their wild-type counterparts [25]. These data further suggest that D66 is a likely substrate of AcrAB-TolC. However, specific genetic and molecular targets of D66 remained unknown.

In this study, we utilized two independent genetic approaches that had not previously been attempted with D66 to identify bacterial genes that affect D66 activity and potentially encode targets of the compound. The results reveal an unexpected phenotype for strains with mutations in *gmhB*, which encodes a cytoplasmic protein that contributes to the biosynthesis of the core of LPS. The data also confirm the role of outer membrane LPS and efflux pumps in excluding small molecules from the cell, but they do not identify a potential molecule target for D66.

## 2. Materials and Methods

### 2.1. Strains, Media, and Antibiotics

Strains included are listed (Table 1). Unless otherwise stated, bacterial cultures were grown in LB [30] at 37 °C) with aeration. Low-phosphate, low-magnesium medium (LPM) at pH 5.5 (LPM 5.5) was made as follows: 5 mM KCl (Fisher Scientific, Pittsburgh, PA, USA), 7.5 mM (NH_4_)_2_SO_4_ (Fisher Scientific), 0.5 mM K_2_SO_4_ (Fisher Scientific), 10 mM glucose (Fisher Scientific), 49 µM MgCl_2_ (Fisher Scientific), 337 µM PO_4_^−^ (Fisher Scientific), 0.05% casamino acids (Fisher Scientific), 80 mM MES (Fisher Scientific), adjusted to pH 5.5 with 5 M NaOH (Fisher Scientific) and sterile-filtered. LPM at pH 7.0 (LPM 7.0) was buffered with 80 mM Tris-HCl (Fisher Scientific) and adjusted to pH 7.0 with 5 M NaOH, then sterile-filtered. Concentrations of antibiotics used when applicable: kanamycin (Kan; Technova, Madison, WI, USA; 30 µg/mL); chloramphenicol (Cm; Fisher Scientific; 34 µg/mL); streptomycin (Str; Fisher Scientific; 30 µg/mL). Novobiocin (Sigma–Aldrich, Saint Louis, MO, USA) and erythromycin (Sigma–Aldrich) were used at various concentrations, as described in Results.

### 2.2. Evolution of D66-Resistant E. coli K12 ΔtolC Mutants and Genetic Analyses

An overnight culture derived from a single colony was separated into six independent cultures that were serially passaged in the presence of increasing concentrations of D66 (MolPort, Riga, Latvia; 002-888-832; 90% purity; Clc1cc(Cl)cc(NC(=O)N2CCCN(Cc3ccccc3)CC2)c1) starting at 1/4X the MIC (47 ± 1 μM) and increasing stepwise in ¼ increments to 3X MIC (141 μM; the solubility limit is approximately 150 μM). In parallel, the *E. coli* K12 *ΔtolC* parent strain was serially passaged in two independent cultures in LB containing 2% DMSO, the vehicle for D66. Cultures were grown at 37 °C with aeration until the OD_600_ reached at least ~0.5, as determined visually, and then serially passaged at a dilution of 1:100 into LB containing stepwise increases of D66. When growth at 3 × MIC D66 was achieved (12 passages), isolates were recovered on LB agar and tested for heritable resistance in LB with 3 × MIC D66. Each bacterial culture was pelleted, and genomic DNA from overnight LB cultures was extracted by SeqCenter (Pittsburgh, PA, USA). Library preparation and whole genome sequencing were performed using Illumina WGS by SeqCenter. The genomes of the six independent D66-resistant clones and the two parental DMSO-exposed clones were compared. Variants were called using Breseq [35].

### 2.3. Construction of E. coli K12 and S. Typhimurium SL1344 gmhB Mutant Strains

Gene deletions were constructed using the standard Lambda Red recombination method [36]. The forward primer was designed to bind 50 base pairs (bp) upstream of *gmhB* (1. *gmhB* LR FWD; Table 2). Reverse primers were designed to bind 50 bp downstream of the stop codon (2. *gmhB* LR Del REV) or 50 bp downstream of bp 243 within *gmhB* for the *∆81gmhB* truncation mutants (3. *gmhB* LR Trunc REV). PCR-generated dsDNA product was produced using a Kan resistance cassette template and transformed into SL1344. Following transformation, cells were incubated at 37 °C in SOC for 4 h to allow for recombination and plated on LB-Kan agar plates. Single colonies were picked for sequencing to verify the deletion.

### 2.4. Plasmid Construction for Complementation of gmhB

The complementing pBAD33-*gmhB* plasmid was constructed by amplifying the pBAD33 plasmid (primers 4. pBAD33 amplification FWD and 5. pBAD33 amplification REV). The *gmhB* gene was amplified from genomic K12 *E. coli* DNA (primers 6. *gmhB* gene amplification FWD and 7. *gmhB* gene amplification REV). Linear PCR products were recombined using NEB Gibson Assembly^®^ Master Mix (New England Biolabs, Ipswich, MA, USA), transformed into K12 *E. coli*, and plated to LB-Kan, Cm agar plates. Plasmids were extracted from single colonies and sequenced for verification.

### 2.5. Sensitivity to Novobiocin and Erythromycin

Overnight cultures of the indicated strains were grown in LB at 37 °C with aeration. Cultures were diluted 1:100 to an approximate OD_600_ of 0.01 in fresh LB and distributed into a flat-bottomed polystyrene 96-well plate. Either novobiocin (dissolved in water), D66 (dissolved in DMSO), or erythromycin (dissolved in ethanol) was immediately added to the desired final concentration. DMSO was kept to a final concentration of 1%. The plate was cultured at 37 °C with shaking for 18 h, and the final OD_600_ values were measured using a BioTek Synergy H1 plate reader (Agilent BioTek Synergy H1 plate reader (Agilent, Santa Clara, CA, USA).

### 2.6. Selection of Transposon Mutants with Reduced Fitness in the Presence of D66

An aliquot of an *S.* Typhimurium transposon mutant library [34,37] was diluted 1:100 in 66 mL LB and grown to an OD_600_ of 0.3 for approximately 4 h at 37 °C with aeration. Aliquots of 11 mL were distributed to six 100 mL flasks and treated with vehicle (water) and polymyxin B (Sigma–Aldrich; 0.5 μg/mL) and/or with vehicle (DMSO) or D66 (25, 50, or 100 μM). Samples were incubated for 24 h at 37 °C with aeration and then diluted to an OD_600_ of 0.3 with or without polymyxin B in the presence of DMSO or the compound. Samples were diluted to an OD_600_ of 0.3 in the same medium and grown for another 24 h until 48 h after the initial treatment. Methods for library DNA preparation, sequencing, and data analysis were previously described [38]. In brief, bacteria were pelleted and lysed, and the lysate was used as a template for PCR using primers directly flanking the N_18_ barcode. The frequency of each barcode was determined by Illumina sequencing of at least 25 bases. The aggregated abundances for the input and output libraries were statistically analyzed using DESeq2 [39], and the log_2_-fold changes and FDRs were reported. D66-treated samples were compared with DMSO controls to assess the loss of transposon mutants over 48 h. Three biological replicates were performed.

### 2.7. Validation of Transposon Mutants in Competitive Fitness Assays

The 14028’s parent (wild type) strain marked with Kan or Cm was grown in competition with strains harboring deletions of genes of interest (*dgkA*, *rfe*, *barA*, or *dedA*) marked with Kan (sense direction) or Cm (antisense direction) [40]. Overnight cultures in LB medium were diluted 1:100 in LPM 5.5 medium and then treated with 45 μM D66. Cultures were sampled at 0, 24, and 48 h and plated on LB-Str (enabling all bacteria to grow), LB-Str-Kan, and LB-Str-Cm (selecting for the parent or mutant strain). The ratio of wild type to total CFU per mL was calculated. Three biological replicates were performed for each WT-Kan versus mutant-Cm and each WT-Cm versus mutant-Kan experiment.

### 2.8. Statistical Analyses

Prism version 10.11 (GraphPad Software, Boston, MA, USA) version 10.11 was used for statistical comparisons.

## 3. Results

### 3.1. Mutations in gmhB/yaeD Confer Heritable D66 Resistance in Broth

Protein targets of small molecules with anti-infective activity at the bacterial cell membrane have been identified by selection for resistant mutants, followed by whole genome sequencing [41,42]. Since D66 does not inhibit bacterial growth under normal broth conditions, we screened for resistant mutants in strain backgrounds that sensitize *S.* Typhimurium and *E. coli* to D66. Selection for resistant mutants was previously carried out in an *S.* Typhimurium strain lacking the *acrAB* locus, which encodes two subunits of the AcrAB-TolC efflux pump [25]. However, when screening for resistant mutants in this background, we exclusively identified mutants that increase the expression of multiple TolC-dependent efflux pumps, potentially because TolC is a subunit of multiple efflux pumps [25,43,44]. In this study, we selected D66-resistant isolates in an *E. coli ΔtolC* strain background. D66 has a minimum inhibitory concentration (MIC) of 47 ± 1 µM in an *E. coli ΔtolC* strain [25] (Figure 1A). We found that five of six independent isolates that grew in the presence of stepwise increases of D66 (up to 3 times the MIC) had non-synonymous mutations in *gmhB* (Figure 1B). The sixth isolate had a mutation in *ygeG*, encoding an uncharacterized putative chaperone that we did not pursue further (Table 3).

GmhB is a globular cytoplasmic protein (Figure 1C) that facilitates the biosynthesis of the inner core region of LPS; it is a D,D-heptose 1,7-bisphosphate phosphatase that converts D-glycero-β-D-manno-heptose 1,7-bisphosphate to ADP-l-glycero-β-D-manno-heptose [45]. Of the five resistant isolates with mutations in *gmhB*, one isolate had a mutation just after the N-terminal domain that binds magnesium and the enzymatic substrate (gray line and arrow in B and C, respectively). The remaining four mutations were near the start of the catalytic domain (green lines and arrows in B and C, respectively). To establish whether the presence of the N-terminal domain contributed to phenotypic resistance to D66, we constructed strains with chromosomal deletions of the entire open reading frame (*ΔgmhB*) and strains lacking the first 81 amino acids of GmhB (*Δ81gmhB).* In the *E. coli ΔtolC* mutant background, both deletions conferred resistance to D66-mediated growth inhibition, and complementation with full-length *gmhB* restored susceptibility to D66 at 2X MIC (Figure 1D). It thus appears that *E. coli ΔtolC* strains lacking *gmhB* are resistant to D66.

To determine whether resistance to D66 following the loss of *gmhB* was unique to *E. coli* and/or dependent on the *ΔtolC* mutation in the parent *E. coli* strain, we constructed *gmhB* mutations in a wild-type *S.* Typhimurium background. A sub-growth-inhibitory concentration of polymyxin B (0.5 μg/mL) was utilized to damage the outer membrane and facilitate D66 access to the inner membrane [25]. In the presence of polymyxin B, exposure of wild type to increasing concentrations of D66 revealed sensitivity to the compound, whereas the *ΔgmhB* and *Δ81gmhB* strains remained resistant to D66 (Figure 1E). Complementation with the *pgmhB* plasmid restored sensitivity. These data do not support the idea that resistance to D66 results from an interaction between D66 and the N-terminus or any other domain of GmhB. Instead, they show that loss of *gmhB* confers resistance to D66 in *E. coli* and *S.* Typhimurium.

### 3.2. Strains Lacking gmhB Have a Less Permeable Cell Envelope

We hypothesized that the loss of *gmhB* conferred resistance to D66 by decreasing bacterial cell envelope permeability. Therefore, we established whether *ΔgmhB* mutants showed increased resistance to the antibiotic novobiocin, which has limited access to its cytosolic target LptB in the presence of a functional cell envelope [46,47]. *E. coli* strains lacking *gmhB* were more resistant to novobiocin than the parent *ΔtolC* strain (Figure 2A). To determine whether resistance in *ΔgmhB* strains extended to other antibiotics, we employed erythromycin, which, like novobiocin, has difficulty crossing an intact outer membrane [48]. In *E. coli*, the *ΔgmhB* strain showed increased resistance to erythromycin compared with the wild type and the complemented *gmhB* strain (Figure 2B). It was possible that the *E. coli ΔtolC* background confounded these observations and/or that loss of *gmhB* only confers antibiotic resistance in *E. coli*. Therefore, we monitored novobiocin resistance in an *S.* Typhimurium wild-type background with sub-inhibitory concentrations of polymyxin-B, which allows novobiocin to cross the outer membrane [49]. The wild-type *S.* Typhimurium and complemented *ΔgmhB* strains were sensitive to novobiocin, whereas the *ΔgmhB* strain showed no growth inhibition at concentrations of up to 40 ug/mL (Figure 2C). These results suggest that *ΔgmhB* resistance to D66 in *E. coli* and *S.* Typhimurium reflects a general decrease in outer membrane permeability.

### 3.3. Competition for Growth Identifies Transposon Mutants with Reduced Fitness in the Presence of D66

To gain a deeper understanding of the effects of D66 on the bacterial cell, clones from a transposon mutant library with reduced growth in the presence of D66 were identified. A decrease in fitness in the presence of D66 may indicate that the corresponding genes are required for defense against the compound and thereby suggest a potential target. We exposed an *S.* Typhimurium transposon library to D66 in LB containing 0.5 μg/mL polymyxin B to enhance outer membrane permeability and increase D66 access to bacteria [37] (Figure 3A). As expected, clones with reduced fitness included those containing insertions in genes encoding AcrAB-TolC efflux pump subunits (Appendix A), further supporting the idea that D66 is a substrate of this pump. Beyond efflux pumps, insertions that correlated with reduced fitness were identified in four genes (*barA*, *rfe*, *dgkA*, and *dedA*) involved in different aspects of cell envelope stability [50,51,52,53]. Competitive fitness of gene deletion strains marked with kanamycin (Kan)- or chloramphenicol (Cm) was tested in direct pairwise competition with the Cm- or Kan-marked wild-type strain. A low-phosphate, low-magnesium medium at pH 5.5 (LPM 5.5), which mimics conditions within the macrophage phagosome [54,55] and increases outer membrane permeability [20,25], was employed to facilitate D66 access to the bacterial cell. In LPM 5.5 with D66, all four Kan-marked mutant strains were outcompeted by Cm-marked wild type (Figure 3B), and Cm-marked *∆dgkA*, *∆barA*, and *∆rfe* were outcompeted by Kan-marked wild type (Figure 3C)**.** Loss of *dgkA*, *barA*, *rfe*, and possibly *dedA*, therefore, reduces fitness in the presence of D66.

Given that the four genes (*barA*, *rfe*, *dgkA*, and *dedA*) identified in the Tn screen are associated with cell envelope stability ([50,51,52,53]; see discussion), we sought to determine whether they contribute to the exclusion of other small molecules from the cell. To that end, *S.* Typhimurium strains were grown in LB with increasing concentrations of novobiocin. All strains grew to an OD_600_ > 1.0 at low concentrations of novobiocin, as expected (Figure 3D). While the wild-type strain showed no growth inhibition up to 40 µg/mL of novobiocin, the four mutant strains tested had reduced growth at varying novobiocin concentrations: *∆barA* (22 μg/mL), *∆rfe/wecA* (7 μg/mL), *∆dgkA* (22 μg/mL), and *∆dedA* (22 μg/mL). Increased sensitivity to novobiocin is consistent with a compromised outer membrane barrier, suggesting that the genes identified in the Tn fitness screen enable D66 and other small molecules to accumulate more efficiently within the cell.

## 4. Discussion

D66 reduces bacterial recovery from macrophages and mice, is fairly well-tolerated by mice, and, in broth culture, targets the bacterial cell membrane [25]. These observations indicate that this molecule has anti-infective properties but does not overtly lyse bacterial membranes. In contrast, other small anti-infectives we have studied (JD1, JAV1, and JAV2) are bacteriolytic. While JD1, for instance, lyses bacteria in broth at 10-fold higher concentrations than are needed to promote bacterial killing in macrophages, the bacteriolytic compounds were not further pursued [21,24]. For D66, we reasoned that selecting of resistant mutants at growth-inhibitory concentrations could identify loci that encode or produce a target of the compound. A complementary screen for Tn-insertion clones with reduced fitness at sub-inhibitory concentrations of D66 aimed to identify loci that, when intact, confer resistance to D66 by limiting access to a target or damage caused by D66. However, our genetic screens did not reveal direct targets of D66, suggesting target redundancy, target essentiality, or the existence of non-protein targets such as bacterial cell-membrane lipids or other structural components required in the context of infection. Nevertheless, these results contribute to our understanding of the components of bacterial cell envelopes that limit small-molecule access to cells.

The Tn-library fitness competition identified clones with insertions in genes that contribute to the maintenance of the bacterial cell envelope barrier. For instance, the reduced fitness of clones with insertions in genes encoding AcrAB-TolC underscores the likelihood that this efflux pump exports D66 [25]. These data confirm the rationale for screening for D66-resistant mutants in an *E. coli ∆tolC* background, enabling the accumulation of compounds within cells. We speculate that *gmhB* mutations were not previously identified in Enterobacterales screens for resistance to antibiotics because many antibiotics are also exported by RND efflux pumps, especially AcrAB-TolC.

The validation of a *barA* contribution to fitness in the presence of D66 may reflect that the BarA sensor-kinase activates global regulators, including ones that induce *acrAB* expression [56,57,58]. Other clones with reduced fitness in the presence of D66 had Tn insertions in genes that maintain the integrity of the outer membrane barrier. Two of these genes are needed for O-antigen polysaccharide biosynthesis. In the cytosol, WecA/Rfe catalyzes an early step in O-antigen production, the transfer of α-N-acetylglucosaminyl to undecaprenyl-phosphate (UndP) [2]. Loss of *wecA/rfe* results in the production of a truncated LPS and increases bacterial outer membrane permeability [59,60], consistent with our observations. DedA is an integral membrane protein that is a putative UndP flippase predicted to enable UndP recycling [50,61], and strains lacking *dedA* are temperature- and antibiotic-sensitive [62,63,64]. The diacylglycerol kinase encoded by *dgkA* is needed for membrane integrity and the production of glycerophospholipids via the recycling of membrane-associated diacylglycerol (DAG) [65]. Loss of *dgkA* increases sensitivity to antibiotics and polymyxins [52,66]. Overall, the transposon screen implicated the AcrAB-TolC efflux pump, the LPS layer, and membrane lipids, each of which is major contributors to the integrity of the Gram-negative cell envelope barrier that protects bacteria from D66.

In contrast to the Tn-library fitness competition, selection for D66-resistant mutants yielded a result that at first glance appeared counterintuitive: loss of *gmhB*, an LPS inner core phosphatase, conferred resistance to D66. LPS generally prevents small molecules from crossing the Gram-negative outer membrane, and mutations in LPS core biosynthesis genes are typically expected to confer sensitivity, not resistance, to small molecules. To resolve this conundrum, we showed that *gmhB* mutant strains are resistant not just to D66 but also to novobiocin and erythromycin, suggesting that loss of *gmhB* results in a more robust, less permeable outer membrane barrier. As a potential mechanism for these observations, we suggest redundancy between GmhB and promiscuous phosphatases, an idea previously proposed for *E. coli*, *Klebsiella pneumoniae*, and *Yersinia pseudotuberculosis* [45,67,68,69]. In *E. coli* (K–12 derivative W3110), unidentified phosphatases are hypothesized to partially compensate for the loss of *gmhB* by adding heptose to the LPS core, producing a mixture of heptose-less and heptose-rich forms of LPS [45]. In *K. pneumoniae*, strains lacking *gmhB* also produced a mixture of truncated and full-length LPS [69]. However, full compensation for loss of *gmhB* by an unidentified promiscuous phosphatase was observed in an *E. coli* (K–12 derivative BW25113) based on the production of an LPS with normal gel mobility [67]. Similarly, in *Y. pseudotuberculosis*, *gmhB* is fully redundant with unknown phosphatases with respect to LPS induction of host cell NF-κB [68]. In sum, unidentified phosphatases appear to compensate to varying degrees for the loss of *gmhB* in Gram-negative bacteria, depending on experimental conditions. Therefore, we speculate that stress induced by incubation with D66, novobiocin, or erythromycin stimulates promiscuous phosphatases in *S.* Typhimurium and *E. coli*, resulting in a more robust outer membrane LPS barrier.

It remains unclear how D66 crosses the LPS layer to reach its target(s) during infection. In macrophages, the compound could gain access to bacteria when host-derived soluble, innate immune molecules, such as cationic antimicrobial peptides, erode the LPS layer of the outer membrane and possibly occupy bacterial efflux pumps, reducing their efficiency. The macrophage infection assay screening platform that identified D66 (SAFIRE, Screen for Anti-infectives using Fluorescence microscopy of IntracellulaR Enterobacteriaceae) may be biased for hydrophobic compounds that require host innate immunity to gain access to bacteria. As an alternative to the SAFIRE platform, future screens for anti-infective compounds could incorporate a primary screen for hits that inhibit bacterial growth under broth conditions that compromise the outer membrane barrier or efflux pumps. Such an approach may identify compounds that interact not only with lipids but also with bacterial membrane proteins. A secondary screen with SAFIRE could single out compounds with anti-infective activity and limited host cell toxicity.

The study of small molecules with targets in the *Mycobacterium tuberculosis* inner membrane provides a framework for thinking about D66: in *M. tuberculosis*, small hydrophobic molecules that target the inner membrane can interact with both lipids and membrane proteins. Unlike classical Gram-negative bacteria, Mycobacteria have a lipophilic outer membrane that enables passive diffusion of hydrophobic or amphiphilic compounds [70,71,72]. A frequent target for growth-inhibitory small hydrophobic molecules in *M. tuberculosis* is MmpL3, a mycolate flippase required for mycolate transport to the outer membrane [73,74]. Specifically, MmpL3 is a potential target for the pyrrole derivative BM212, as demonstrated by the selection of BM212-resistant mutants [42]. BM212 competitively binds soluble MmpL3 (49), but whether mutations in MmpL3 conferring resistance reduce binding was not tested (19, 49). BM212 also directly binds at least one other membrane protein, but the relevance of these “off-target” interactions is not clear [75]. However, a series of small-molecule anti-infectives that competitively bind MmpL3, inhibiting mycolate transport, have MmpL3-dependent bacteriolytic activity [76]. Derivatives of this series were not toxic to mammalian cells, showing efficacy in a murine model of acute tuberculosis [77]. Critically, these data suggest that hydrophobic molecules targeting the bacterial inner membrane can interact with both lipids and membrane proteins. This is an intriguing concept, as the potential for pleiotropic effects with hydrophobic small molecules may slow the rise of genetic resistance and thwart the emergence of tolerance. Thus, it may be desirable for small hydrophobic molecules to have multiple mechanisms of action and/or targets, provided host toxicity is minimal. For these reasons, establishing the molecular targets of small-molecule anti-infectives remains an important goal.

## 5. Conclusions

The current study attempted to use two complementary genetic approaches to identify bacterial genes that affect D66 activity and encode or produce molecular targets of the compound. While this goal was not met, the work revealed an unexpected phenotype for strains with mutations in the LPS biosynthesis gene *gmhB*. In addition, the data reiterate the role of outer membrane LPS and RND efflux pumps in excluding small molecules from the Gram-negative cell.

## Figures and Tables

**Figure 1 microorganisms-13-01521-f001:**
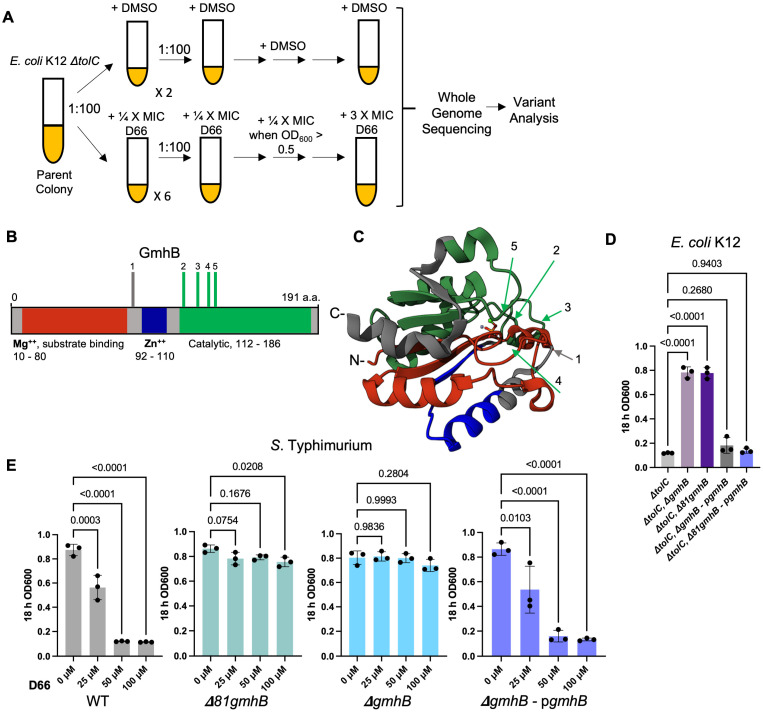
Loss of LPS-biosynthesis gene *gmhB* in *E. coli* and *S.* Typhimurium confers resistance to D66. (**A**) Selection scheme for identifying resistant isolates. All samples originated from a single *E. coli* K12 *∆tolC* parent colony grown in LB overnight and diluted 1:100 into 2 mL of LB containing DMSO (2 biological replicates) or D66 (6 biological replicates). The concentration of D66 was increased by 12.5 μM (¼ × MIC) at each passage. DMSO controls were transferred at the same time as the treated samples. Variant analysis was performed with the BreSeq program, and differences compared with the DMSO controls. An estimated 80–86 generations were needed to reach resistance (3 × MIC), based on time of transfer and ~30 min doubling time. (**B**,**C**) Domain map and 3D structure of GmhB. Red indicates the N-terminal amino acid magnesium and substrate-binding domain, blue the zinc-binding domain, and green the catalytic domain. Vertical bars (**A**) and arrows (**B**,**C**) show the locations of independent mutations that confer resistance to D66 in *E. coli* K12 *∆tolC*. (**D**) *E. coli* K12 *ΔtolC* sensitivity to D66 is lost with deletion of *gmhB* (*∆gmhB*) or of the first 81 amino acids of *gmhB* (*∆81gmhB*). Sensitivity was restored upon complementation (*∆gmhB * + *pgmhB* and *∆81gmhB* + *pgmhB*). The strains indicated were grown overnight in LB and inoculated 1:100 in LB with DMSO or D66 (100 μM). Absorbance was measured after 18 h. Mean and SEM are shown. One-way ANOVA with Tukey’s multiple comparison posttest, *n* = 3 biological replicates. (**E**) *S.* Typhimurium SL1344 sensitivity to D66 in the presence of PMB is lost with deletion of the *gmhB* N-terminus or the entire open reading frame (*∆81gmhB* and *∆gmhB*, respectively). Sensitivity was restored upon complementation (*∆gmhB* + *pgmhB*). The strains indicated were grown overnight in LB and inoculated 1:100 in LB supplemented with 0.5 μg/mL PMB to weaken the outer membrane barrier. Three doses of D66 were added, and absorbance was measured after 18 h. Mean and SEM are shown. One-way ANOVA with Tukey’s multiple comparison posttest, *n* = 3 biological replicates.

**Figure 2 microorganisms-13-01521-f002:**
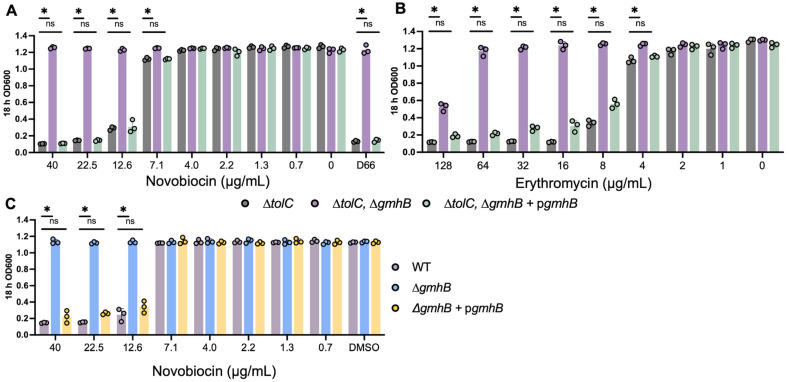
Loss of *gmhB* confers resistance to antibiotics that is reversed by complementation with *gmhB*. (**A**,**B**) *E. coli* K12 *ΔtolC* sensitivity to novobiocin or erythromycin. The strains indicated were grown overnight in LB and inoculated 1:100 in LB with DMSO, D66 (100 μM), or the indicated concentration of novobiocin (**A**) or erythromycin (**B**). Absorbance was measured after 18 h. Mean and SEM are shown. Two-way ANOVA with Tukey’s multiple comparison posttest, *n* = 3 biological replicates. * = *p* < 0.001; ns = not significant. (**C**) *S.* Typhimurium SL1344 sensitivity to novobiocin in the presence of PMB. The strains indicated were grown overnight in LB and inoculated 1:100 in LB with DMSO or the indicated concentration of novobiocin. Absorbance was measured after 18 h. Mean and SEM are shown. Two-way ANOVA with Tukey’s multiple comparison posttest, *n* = 3 biological replicates. * = *p* < 0.0001; ns = not significant.

**Figure 3 microorganisms-13-01521-f003:**
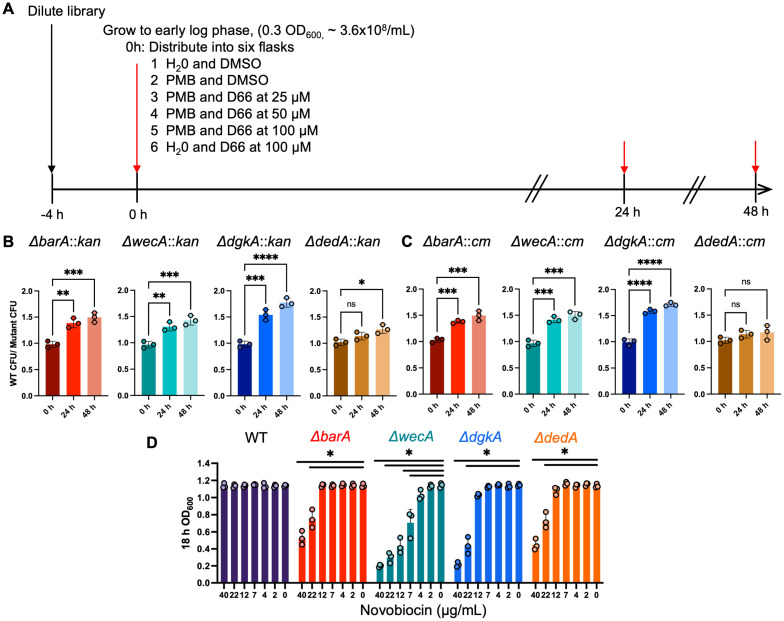
Tn-mutant screen schematic and analysis of corresponding deletion mutants. (**A**) Selection screen with the *S.* Typhimurium Tn-mutant library in the presence of D66. Schematic of the growth of the Tn-mutant library under 6 conditions tested. The Tn mutant library was diluted 1:100 and grown to an OD_600_ of ~0.3. Cultures were distributed into 6 flasks with either 0.5 µg/mL PMB and D66 or vehicle controls. Samples were collected at t_0_, t_24_, and t_48_ (red arrows), and DNA from the pellets was sequenced. (**B**,**C**) Competitive fitness assays comparing the *S.* Typhimurium *14028s* wild-type parent strain marked with Cm or Kan to mutant strains marked with Kan (**B**) or Cm (**C**). Overnight cultures were inoculated 1:100 into LPM 5.5 medium, grown for 24 h, and colony-forming units (CFU) were enumerated on Kan and Cm agar plates. The mixed cultures were diluted again, grown for another 24 h, and enumerated as before. The mean ratio of wild type to mutant CFU with SEM for three biological replicates is shown. One-way ANOVA, with a Tukey’s multiple comparisons posttest, *n* = 3 biological replicates. * = *p*-value < 0.05, ** = *p*-value < 0.01, *** = *p*-value < 0.001, **** = *p*-value < 0.0001; ns = not significant. (**D**) Sensitivity of mutants to novobiocin. Overnight cultures of *S.* Typhimurium 14028s wild type and the indicated gene deletion strains were diluted 1:100 into LB containing novobiocin. Absorbance was measured at 18 h. Mean and SEM are shown. One-way ANOVA, with a Tukey’s multiple comparisons posttest, *n* = 3 biological replicates. * = *p*-value < 0.001.

**Table 1 microorganisms-13-01521-t001:** Strain list.

Reference #	Strain Name	Species	Antibiotic	Source
ALR1247	K–12 derivative BW25113	*E. coli*		[31]
JLD1285	K–12 *ΔtolC*	*E. coli*	Kan	[32]
SCA1496	K12 *ΔtolC*, *ΔgmhB*	*E. coli*	Kan	This Study
SCA1497	K12 *ΔtolC*, *Δ81gmhB*	*E. coli*	Kan	This Study
SCA1544	K12 *ΔtolC*, *ΔgmhB-pBAD33-gmhB*	*E. coli*	Kan, Cm	This Study
SCA1547	K12 *ΔtolC*, *Δ81gmhB-pBAD33-gmhB*	*E. coli*	Kan, Cm	This Study
CSD001	SL1344	*S.* Typhimurium	Str	[33]
SCA1498	SL1344 *ΔgmhB*	*S.* Typhimurium	Str, Kan	This Study
SCA1499	SL1344 *Δ81gmhB*	*S.* Typhimurium	Str, Kan	This Study
SCA1546	SL1344 *ΔgmhB-pBAD33-gmhB*	*S.* Typhimurium	Str, Kan, Cm	This Study
CSD1347	14028s; MZ0431	*S.* Typhimurium	Str	[34]
CSD1351	14028s:CM; MZ2758	*S.* Typhimurium	Str, Cm	[34]
CSD1352	14028s:KAN; MZ2770	*S.* Typhimurium	Str, Kan	[34]
SCA1375	14028s STM14_2915; *ΔdedA*	*S.* Typhimurium	Kan	[34]
SCA1373	14028s STM14_4716; *Δrfe*	*S.* Typhimurium	Kan	[34]
SCA1369	14028s STM14_3566; *ΔbarA*	*S.* Typhimurium	Kan	[34]
SCA1371	14028s STM14_5093; *ΔdgkA*	*S.* Typhimurium	Kan	[34]
SCA1376	14028s STM14_2915; *ΔdedA*	*S.* Typhimurium	Cm	[34]
SCA1374	14028s STM14_4716; *Δrfe*	*S.* Typhimurium	Cm	[34]
SCA1370	14028s STM14_3566; *ΔbarA*	*S.* Typhimurium	Cm	[34]
SCA1372	14028s STM14_5093; *ΔdgkA*	*S.* Typhimurium	CM	[34]

**Table 2 microorganisms-13-01521-t002:** Primer list.

#	Primer	Primer Sequence
1	*gmhB* LR FWD	5′GTGTAGGCTGGAGCTGCTTCTCGAAACATGCGATACTAGCGTCACATGCCTTATTAAGGAGCTATAAAAG
2	*gmhB* LR Del REV	5′AGGAAGACAAGCGGAAAAATGCATTTTTATTTCAACCGCTCATCTTTTAAATGGGAATTAGCCATGGTCC
3	*gmhB* LR Trunc REV	5′CCCTGCGGATGATGCGGGCAATAATAGATACCATCCAGATCGACATCTCGATGGGAATTAGCCATGGTCC
4	pBAD33 amplification FWD	5′GCAAAAACCGGCACAATGATCTGATTCGTTACCAATTATGACAACTTGACGGCTACAT
5	pBAD33 amplification REV	5′CTCTTCGCATAAACCTGATTGGAAGATCGGGCTCGCC
6	*gmhB* gene amplification FWD	5′AGTGGCGAGCCCGATCTTCCAATCAGGTTTATGCGAAGAGCACTT
7	*gmhB* gene amplification REV	5′CATAATTGGTAACGAATCAGATCATTGTGCCGGTTTTTGCTG

**Table 3 microorganisms-13-01521-t003:** Independent *E. coli ΔtolC* isolates resistant to D66.

Strain	Gene	Nt Position	Mutation	Effect
Mutant 1	*gmhB*	223,075	TTC → ATT	Premature stop
Mutant 2	*gmhB*	223,174	CAC → CGC	His to Arg
Mutant 3	*gmhB*	223,184	CTTTT → CTTA	Frameshift
Mutant 4	*gmhB*	223,192	GCA → GGA	Ala to Gly
Mutant 5	*gmhB*	223,200	TAT → TTT	Tyr to Phe
Mutant 6	*ygeG*	2,991,548	CT → CTT	Frameshift

## Data Availability

The original contributions presented in this study are included in the article/Appendix A. Further inquiries can be directed to the corresponding author.

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
