# Peer review of "Mutations in Genes with a Role in Cell Envelope Biosynthesis Render Gram-Negative Bacteria Highly Susceptible to the Anti-Infective Small Molecule D66"

_microorganisms, 2025, doi:10.3390/microorganisms13071521_

Round 1
Reviewer 1 Report
Comments and Suggestions for Authors
This is a well executed and presented study on the search for the target of a small hydrophobic antimicrobial compound. While the target could not be identified yet, the study offers important insight in the uptake and release of the compound and the role of outer membrane integrity and efflux pumps in this. A few issues:
-Please comment why polymyxinB was used in stead of its nonapeptide derivative to avoid dual effects on both inner and outer membrane permeability and voltage.
-Why were gmhB mutations not identified before in resistance screens for antibiotics like erythromycin?
-Usually, the "real" target can be identified when higher concentrations of the compound are used. Was this tried and unsuccessful?
-A more extensive description of the previous paper (ref 25) and the novelty of the current study should be given. There appears to be some overlap...
-To characterise the effect of D66 further, it is advised to look at synergy with compounds known to have distinct effects on the inner membrane.
Reviewer 2 Report
Comments and Suggestions for Authors
The research article “Mutations in Genes with a Role in Cell Envelope Biosynthesis 2 Render Gram-negative Bacteria Highly Susceptible to the Anti-3 infective Small Molecule D66” is devoted to studying the antibacterial action of the compound D66 against the pathogen Salmonella enterica.
Minor comments:
- The section “2. Materials and Methods”:
- a) indicate the origin of all reagents (manufacturer);
- b) indicate the origin of all antibiotics (manufacturer);
- c) indicate the origin of all bacterial strains;
- d) indicate the origin of compound D66 (manufacturer, purity, etc.).
- The subsection “Sensitivity to novobiocin and erythromycin”, line – 137: BioTek Synergy H1 plate reader - Agilent BioTek Synergy H1 plate reader (Agilent, Santa Clara, CA, USA).
- What molecular targets do the authors think could be for compound D66?
- After the section "4. Discussion", it is necessary to insert the section "Conclusion". In this section, describe the novelty of the research and the main achievements of the research.
- The manuscript contains several minor errors and inaccuracies that the authors can correct.
Despite minor flaws, this article is essential in the field of combating pathogenic microorganisms and overcoming drug resistance.
Reviewer 3 Report
Comments and Suggestions for Authors
Dear authors,
The article is valuable and has a wide audience of readers. It can be published after making minor revisions:
1. The sources of the antibiotics used in the study must be specified.
2. Adding references in the Results section is not in line with standard research writing conventions. If the references support the methodology, please move them to the Methods section. The Results section should focus only on presenting the findings.
3. Kindly remove the word “Figure” from within the figure itself and retain only the figure caption.
